# SUMMIT: An integrative approach for better transcriptomic data imputation improves causal gene identification

Zichen Zhang [1], Ye Eun Bae [1], Jonathan R. Bradley[1], Lang Wu [2] & Chong Wu [3] ✉

Genes with moderate to low expression heritability may explain a large proportion of complex trait etiology, but such genes cannot be sufficiently captured in conventional transcriptome-wide association studies (TWASs), partly due to the relatively small available reference datasets for developing expression genetic prediction models to capture the moderate to low genetically regulated components of gene expression. Here, we introduce a method, the Summary-level Unified Method for Modeling Integrated Transcriptome (SUMMIT), to improve the expression prediction model accuracy and the power of TWAS by using a large expression quantitative trait loci (eQTL) summary-level dataset. We apply SUMMIT to the eQTL summary-level data provided by the eQTLGen consortium. Through simulation studies and analyses of genome-wide association study summary statistics for 24 complex traits, we show that SUMMIT improves the accuracy of expression prediction in blood, successfully builds expression prediction models for genes with low expression heritability, and achieves higher statistical power than several benchmark methods. Finally, we conduct a case study of COVID-19 severity with SUMMIT and identify 11 likely causal genes associated with COVID-19 severity.

Genome-wide association studies (GWASs) have shown that most disease-associated variants reside in noncoding regions[1–3], raising challenges in biological interpretation and target gene identification[4]. These findings also lead to the hypothesis that many genetic variants can affect complex traits by regulating gene expression levels, which has motivated large-scale expression quantitative trait loci (eQTL) analyses[5–7] and transcriptome-wide association studies (TWASs)[8–13]. TWASs integrate expression reference panels (eQTL studies with matched individual-level expressions and genetic data) with complex trait GWAS results to discover gene-trait associations. First, an expression reference panel is used to learn a per-gene expression prediction model by regressing assayed gene expression levels on *cis*-eQTL genotypes (i.e., single nucleotide polymorphisms (SNPs) within 1

megabase of the gene transcription start site and transcription end site). Second, statistical associations are estimated between predicted gene expression levels for GWAS samples and the trait of interest. TWASs have garnered interest within the human genetics community and have deepened our understanding of the genetic basis of many complex traits[14,15].

Despite these encouraging findings, the size of the expression reference panels primarily determines the number of analyzable genes, and hence the power of TWASs. Analyzable genes are defined as genes with satisfactory gene expression prediction models (i.e., prediction accuracy $R^2 \geq 0.01$). For example, building expression prediction models with Genotype-Tissue Expression (GTEx) project v7p data yielded more than twice as many prediction models (i.e., analyzable

[1]Department of Statistics, Florida State University, Tallahassee, FL, USA. [2]Cancer Epidemiology Division, Population Sciences in the Pacific Program, University of Hawaii Cancer Center, University of Hawaii at Manoa, Honolulu, HI, USA. [3]Department of Biostatistics, The University of Texas MD Anderson Cancer Center, Houston, TX, USA. ✉e-mail: cwu18@mdanderson.org

genes) than were developed using GTEx v6p data[16]. For whole blood tissue, the number of analyzable genes increased from 2057 to 6006 solely owing to the increase in the size of the expression reference panel (from 338 samples[17] to 369 samples[16]). Others have also observed that the number of analyzable genes can be significantly increased when using a slightly larger expression reference panel. For example, Zhou et al.[13] show that among the 44 overlapping tissues in GTEx, the average number of analyzable genes increased from 4,570 (v6p) to 7,213 (v8) for one popular TWAS method PrediXcan[8] when the average sample size increased from 160 (v6p) to 332 (v8). More importantly, perhaps due to the small sample sizes of available expression reference panels, the current standard practice of TWASs is to only analyze genes with model performance $R^2 \geq 0.01$[8,9,11]. This practice may fail to capture genes with low expression heritability but large causal effect sizes on the trait of interest, as suggested in previous literature[1]. It is of great interest to construct more powerful gene expression prediction models, especially for genes with low expression heritability.

One potential approach to improving the power of TWASs is to combine individual-level expression reference panel data from several consortia or studies, thereby increasing the sample size of the expression reference panel. While this is straightforward, privacy concerns and subject consent can preclude access to individual-level expression reference panel data, making this approach challenging or practically infeasible. On the other hand, one may use summary-level expression panels (often publicly available) with much larger sample sizes to build expression prediction models. However, to date, there is limited exploration of how one can build expression prediction models using a summary-level expression panel.

In this work, we introduce the Summary-level Unified Method for Modeling Integrated Transcriptome (SUMMIT), a method that integrates summary-level expression reference panel data, derived from much larger sample sizes, with trait GWAS results to identify associated genes for the trait of interest. Specifically, we build gene expression prediction models for blood based on the eQTL summary-level data generated by the eQTLGen consortium[6]. To date, the eQTLGen consortium has conducted the largest meta-analysis involving 31,684 blood samples from 37 cohorts[6], and the corresponding eQTL summary-level data have been made publicly available. Through simulation studies and analyses of GWAS summary statistics from 24 complex traits, we show that SUMMIT improves the accuracy of expression prediction in blood, successfully builds expression prediction models for genes with low expression heritability, and outperforms benchmark methods for identifying risk genes. Additionally, we conduct a case study on COVID-19 severity and identify 11 putatively causal genes.

## Results

### SUMMIT overview

We develop SUMMIT, which extends the conventional TWAS methods[8–12], by leveraging eQTL summary-level data to predict expression levels. SUMMIT consists of three main steps. First, for each gene, we train expression prediction models using a penalized regression framework with eQTL summary-level data (e.g., eQTLGen[6] with sample size of 31,684). Next, we test associations between the predicted gene expression levels and the trait of interest for each fitted expression prediction model with satisfactory performance (e.g., with $R^2 \geq 0.005$). Finally, as $p$-values from different gene expression prediction models can be correlated, we apply the Cauchy combination test[18,19] to aggregate $p$-values from the fitted prediction models and the combined $p$-value from the Cauchy combination test effectively quantifies the overall gene-trait associations. The Cauchy combination test is a computationally efficient $p$-value combination method that provides an accurate $p$-value approximation for highly significant results (which are of interest) and does not require the correlation structure among the combined $p$-values to be estimated.

### Simulation results

In the simulation studies, we first evaluated the accuracy of the expression imputation models generated by SUMMIT and benchmark methods and the corresponding statistical power. Next, we studied the impact of sample size on expression prediction accuracy and TWAS power. We verified that SUMMIT recovered the information of the individual-level expression reference panel from summary-level data, and the improvement in expression prediction accuracy was adequately translated into a higher power of sequential TWASs Fig. 1.

First, we observed that SUMMIT performed better than two widely used competing methods, TWAS-fusion and PrediXcan, yielding a higher average imputation $R^2$ with respect to different gene expression heritability values ($h_e^2$) and proportions of causal SNPs ($p_{causal}$) (Fig. 2a). When $h_e^2 = 0.01$ and $p_{causal} = 0.2$, the average imputation $R^2$ of 1000 replications was estimated to be 0.693% by SUMMIT, showing 1735% improvement compared with PrediXcan and 305%

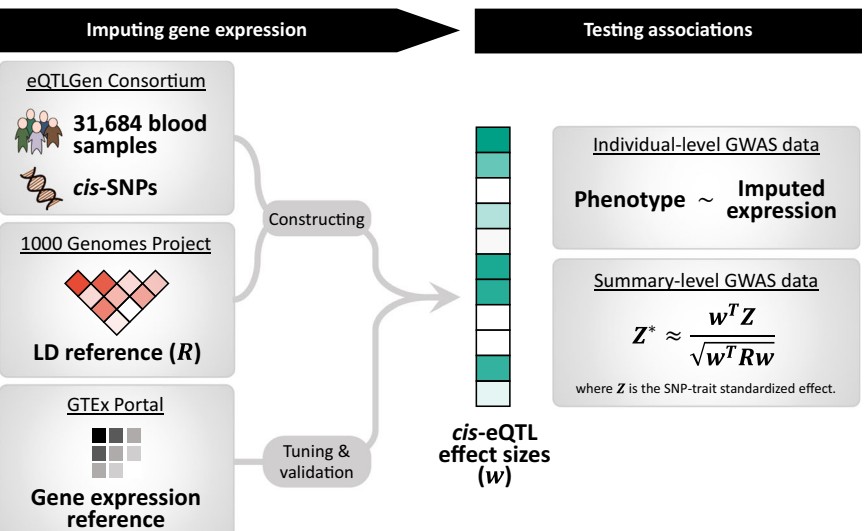

**Fig. 1 | SUMMIT workflow.** SUMMIT consists of three main steps: (1) building prediction models to impute gene expression levels; (2) testing associations between the predicted gene expression levels and the trait of interest; and (3) aggregating results from all fitted prediction models.

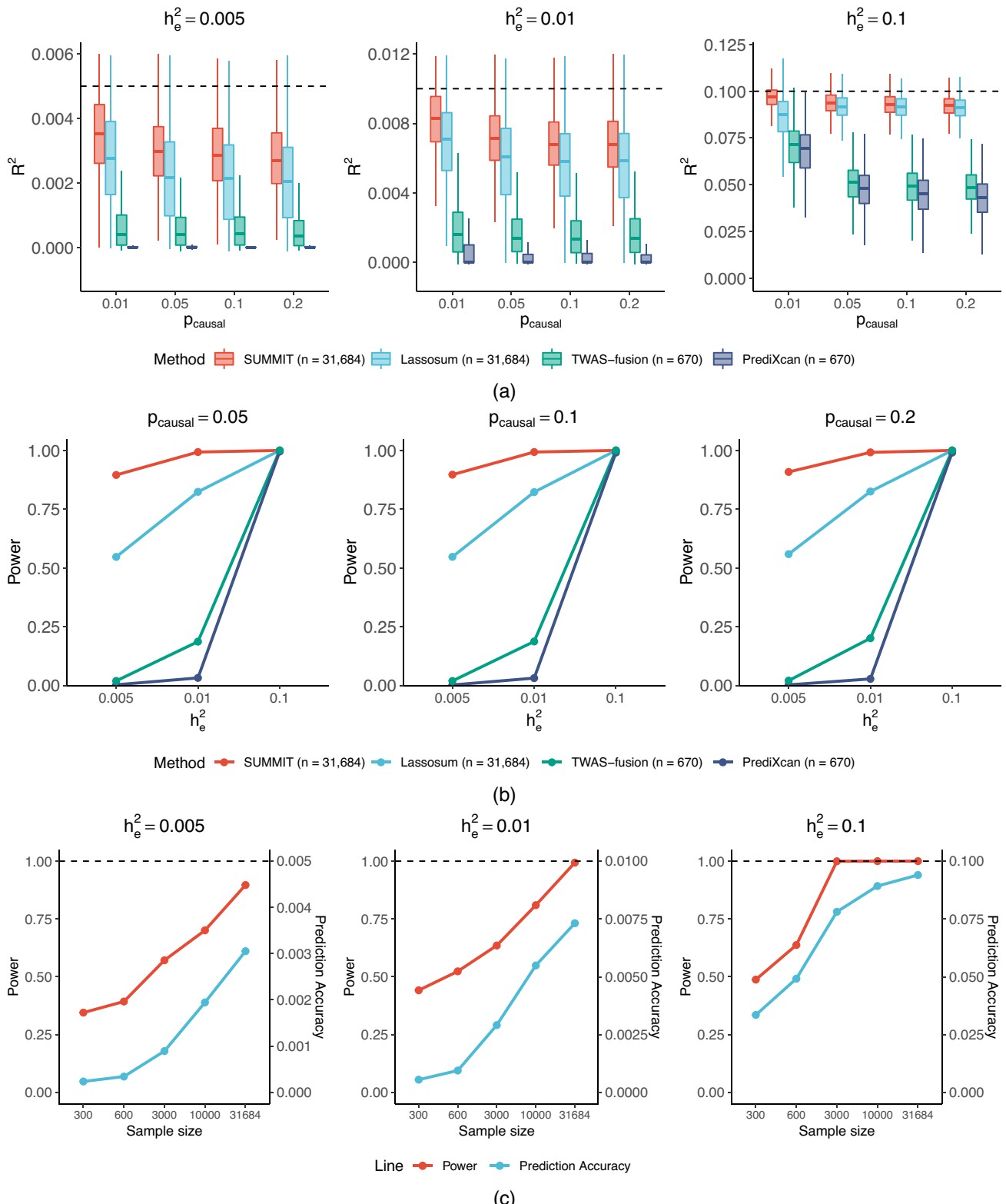

**Fig. 2 | Comparison of performance in simulations using the *CHURC1* gene as an example.** Plots of imputation $R^2$ (**a**) and subsequent power (**b**) in test samples by SUMMIT, Lassosum, TWAS-fusion, and PrediXcan, with varying expression heritability $h_e^2$ and proportion of true causal SNPs $p_{causal}$. The results were based on 1000 simulation replicates. In subfigure (**a**), the box limits represent the lower and upper quartiles, the central line represents the median, and the whiskers represent all samples lying within 1.5 times the interquartile range (IQR). Subfigure (**c**) shows the relationship between the expression panel sample size and TWAS power or expression prediction accuracy using SUMMIT. For subfigures (**b**) and (**c**), we set $h_p^2 = 0.2$; for subfigure (**c**), we set $p_{causal} = 0.05$. *p*-values were calculated by the two-sided tests; empirical power was estimated by the proportions of *p*-values less than the significance threshold $2.5 \times 10^{-6}$. The empirical power comparisons for $h_p^2 \in (0.1, 0.5, 0.8)$ are shown in Supplementary Fig. 1.

improvement compared with TWAS-fusion. Importantly, such improvements in the expression prediction models result in consistently higher TWAS power under different sparsity levels (Fig. 2b). As a note, TWAS power is defined as the discovery rate of associations between predicted expression levels and phenotypic outcomes using simulated independent GWAS data. When $h_e^2 = 0.01$ and $p_{causal} = 0.2$, the power of SUMMIT was 0.992 while those of PrediXcan and TWAS-fusion were 0.028 and 0.201, respectively. In addition, we observed that SUMMIT achieved higher average imputation $R^2$ than Lassosum, a pipeline that is also capable of leveraging summary-level data.

The current standard practice of TWASs is to only analyze genes with imputation $R^2 \geq 0.01$ and not consider genes with lower prediction performance (i.e., genes with imputation $R^2$ between 0.005 and 0.01). However, such genes may have larger causal effect sizes on the trait of interest[1]. To evaluate the performance of different methods under low heritability, we simulated data with $h_e^2 = 0.005$. Figure 2a shows that SUMMIT achieved satisfactory performance under these scenarios. When $h_e^2 = 0.005$ and $p_{causal} = 0.2$, SUMMIT estimated the average imputation $R^2$ at 0.29%, which was much higher than the values yielded by TWAS-fusion (0.057%; 401% improvement) and PrediXcan (0.011%; 2460% improvement). This is because SUMMIT leverages summary-level eQTL data with a larger sample size. Furthermore, SUMMIT also achieved higher average imputation $R^2$ than Lassosum because SUMMIT leverages the genetic distance to estimate the LD matrix and combines results from multiple penalties.

Next, we evaluated the impact of the sample size of the expression reference panel (Supplementary Fig. 2). As expected, the imputation $R^2$ increased as the sample size increased. For the setting of $h_e^2 = 0.05$ and $p_{causal} = 0.2$, when the sample size increased from 300 to 31,684, the average imputation $R^2$ increased from 0 to 0.0474, highlighting the advantages of using a larger expression reference panel. Importantly, the imputation models became more stable (i.e., decreased in variance) as the sample size increased. Additionally, we confirmed that the imputation results from SUMMIT (average imputation $R^2$: 0.0469) were highly similar to those from analyses of individual-level data (average imputation $R^2$: 0.0474), confirming that SUMMIT can capture individual-level information from summary-level data.

Finally, we conducted confirmatory simulation studies (Fig. 2c) to verify that the gains in TWAS power came from an improved expression prediction accuracy. We varied $N$ within (300, 600, 3000, 10,000, 31,684), and $h_e^2$ within (0.005, 0.01, 0.1), and we set $h_p^2 = 0.2$ and $p_{causal} = 0.05$. We observed that the TWAS power and prediction accuracy were highly correlated. As the sample size of the expression reference panel increased, the expression prediction models became more accurate, leading to higher TWAS power. Notably, due to the setup (i.e., the two-sample framework) of the simulations, the gains in the sample size of the expression reference panel could only interact with the TWAS power through better prediction models. The results were similar for $p_{causal} = 0.01$ (Supplementary Fig. 7).

To consider the potential impact of genetic architecture, we considered two additional randomly selected genes, and the results were similar (Supplementary Figs 3–6). Furthermore, we ran the simulations 5,000,000 times (5000 runs for each of 1000 computed weights) under the null hypothesis to evaluate the Type 1 error rates, confirming that all methods maintained well-controlled Type 1 error rates (Supplementary Fig. 8).

In summary, these results demonstrate the potential of SUMMIT for building expression prediction models and conducting subsequent association studies, especially for genes with low expression heritability.

## SUMMIT improves the expression imputation accuracy
We compared the accuracy of the expression prediction models developed using SUMMIT and five benchmark methods, Lassosum,

MR-JTI, TWAS-fusion, PrediXcan, and UTMOST for whole blood tissue. We trained the SUMMIT and Lassosum models with eQTLGen summary data, and the other four benchmark methods were trained with GTEx data. For a fair comparison, we compared the number of genes with estimated $R^2 \geq 0.01$ and only focused on genes that appear in the eQTLGen summary data. The $R^2$ for MR-JTI, TWAS-fusion, PrediXcan, and UTMOST, were based on cross validation and were provided by the original authors, and the $R^2$ for SUMMIT and Lassosum were calculated based on the additional subjects in the GTEx version 8 data, who were not included in the meta-analysis of eQTLGen and thus can be viewed as an independent external dataset. Compared with the benchmark methods, Lassosum (8249 genes), MR-JTI (9576 genes), TWAS-fusion (5411 genes), PrediXcan (7512 genes), and UTMOST (7236 genes), SUMMIT developed satisfactory prediction models for more genes (9749 genes with $R^2 \geq 0.01$). Importantly, SUMMIT could build prediction models for the majority (8936 out of 12,230; 73.1%) of genes that could be analyzed by any of the benchmark methods (Fig. 3a). In addition, SUMMIT was able to establish prediction models of 1836 additional genes that were ignored by benchmark methods that leveraged individual-level data, showing consistent improvement by using a large training dataset. Furthermore, compared with Lassosum, SUMMIT achieved marginally higher prediction accuracy in different quantiles ($T \approx 0.017$ and $p \approx 0.077$, by one-sided Kolmogorov-Smirnov test). Compared with the other four benchmark methods, SUMMIT achieved significantly higher prediction accuracy in different quantiles (MR-JTI: $T \approx 0.080$ and $p < 2.2 \times 10^{-16}$; PrediXcan: $T \approx 0.089$ and $p < 2.2 \times 10^{-16}$; TWAS-fusion: $T \approx 0.240$ and $p < 2.2 \times 10^{-16}$; and UTMOST: $T \approx 0.076$ and $p < 2.2 \times 10^{-16}$; all by one-sided Kolmogorov-Smirnov test).

## SUMMIT identifies more associations than competing methods
To evaluate the performance in identifying significant associations, we applied SUMMIT to the GWAS summary statistics of 24 traits ($N_{total} \approx 5,600,000$ without adjusting for sample overlap across studies, Supplementary Data 1) and compared the results with those of the benchmark methods (for all genes with $R^2 \geq 0.01$). The association results for SUMMIT are summarized in Supplementary Data 1. While SUMMIT analyzed all genes with $R^2 \geq 0.005$ and applied Bonferroni correction accordingly, we focused on the genes with $R^2 \geq 0.01$ for a fair comparison (Fig. 3b). Compared with the benchmark methods, SUMMIT identified more associations for each trait analyzed, showing 50% improvement compared with Lassosum ($T = 334.5$ and $p \approx 0.013$; one-sided by the paired Wilcoxon rank test), 69% improvement compared with MR-JTI ($T = 349$ and $p \approx 0.005$; one-sided), 108% improvement compared with TWAS-fusion ($T = 362$ and $p \approx 0.002$; one-sided), 91% improvement compared with PrediXcan ($T = 335$ and $p \approx 0.005$; one-sided), and 63% improvement compared with UTMOST ($T = 343$ and $p \approx 0.008$; one-sided).

Because different methods test different sets of genes, we also compared the methods over a common set of 3980 genes that could be analyzed by all the methods (Fig. 3c). Again, SUMMIT maintained an edge over the competing methods, showing 16% improvement compared with the second-best-performing method in terms of association pairs identified, Lassosum.

Importantly, SUMMIT was applicable in analyzing genes with low expression heritability ($0.005 \leq R^2 < 0.01$), which have been largely ignored by benchmark methods. Out of the 11,585 genes with $R^2 \geq 0.005$, 1836 had a testing $R^2$ between 0.005 and 0.01. For these 1836 genes, we identified 659 gene-trait associations (Fig. 3b). In comparison, for the remaining 9749 genes, we identified 3339 gene-trait associations, indicating that genes with relatively smaller $R^2$ may be as important as those with larger $R^2$. This finding is in line with the fact that genes with low expression heritability have substantially larger causal effect sizes on complex traits[1].

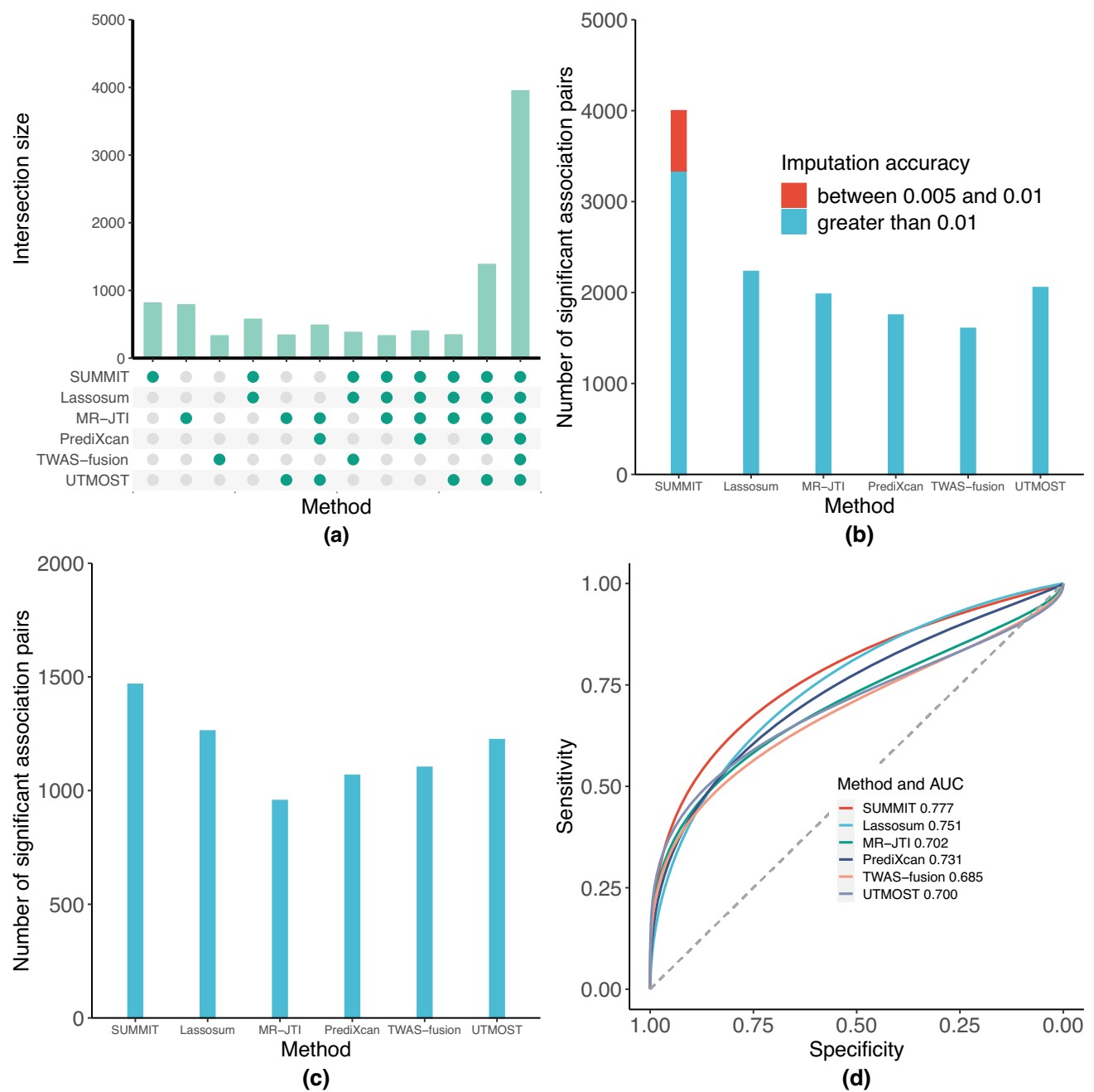

**Fig. 3 | SUMMIT improves the performance of TWASs on real data. a** Is the UpSet plot of overlapping imputation models with $R^2 \geq 0.01$ among different methods. **b** Shows the numbers of associated genes identified by different methods when using all available genes across GWASs of 24 traits, where (**c**) shows the number of associated genes when evaluating a common gene set of all methods. **d** Is the ROC plot for identifying "silver standard" genes.

## SUMMIT achieves higher predictive power for identifying "silver standard" genes

We compared different methods in identifying the likely causal genes that mediate the associations between GWAS loci and traits of interest. Following Barbeira et al.[20], we used a set of 1,258 likely causal gene-trait pairs curated by using the Online Mendelian Inheritance in Man (OMIM) database[21] and a set of 29 gene-trait pairs based on rare variant results from exome-wide association studies[22–24], which provide orthogonal information that is independent of the GWAS results. These genes are counted as "silver standard" genes. Both sets of gene-trait pairs can be found in Supplementary Data 2.

Figure 3d shows that SUMMIT yielded good sensitivity and specificity for identifying the silver standard genes and achieved the highest

AUC (0.777) among all the methods compared. All methods achieved relatively good sensitivity and specificity, showcasing the potential predictive ability of TWAS-type methods to prioritize putative causal genes. At a Bonferroni-corrected significance threshold of $5.21 \times 10^{-6}$, SUMMIT identified 69 genes in the silver standard gene list, whereas Lassosum, the second-best-performing method in terms of AUC, identified 60 (15% improvement). Again, perhaps due to the increase in the sample size of the expression reference panel, the methods based on the summary-level expression reference panel (i.e., SUMMIT and Lassosum) achieved a higher AUC than methods based on the individual-level expression reference panel. In summary, perhaps due to the improvement in the expression prediction models, SUMMIT achieved higher predictive power in terms of prioritizing likely causal genes.

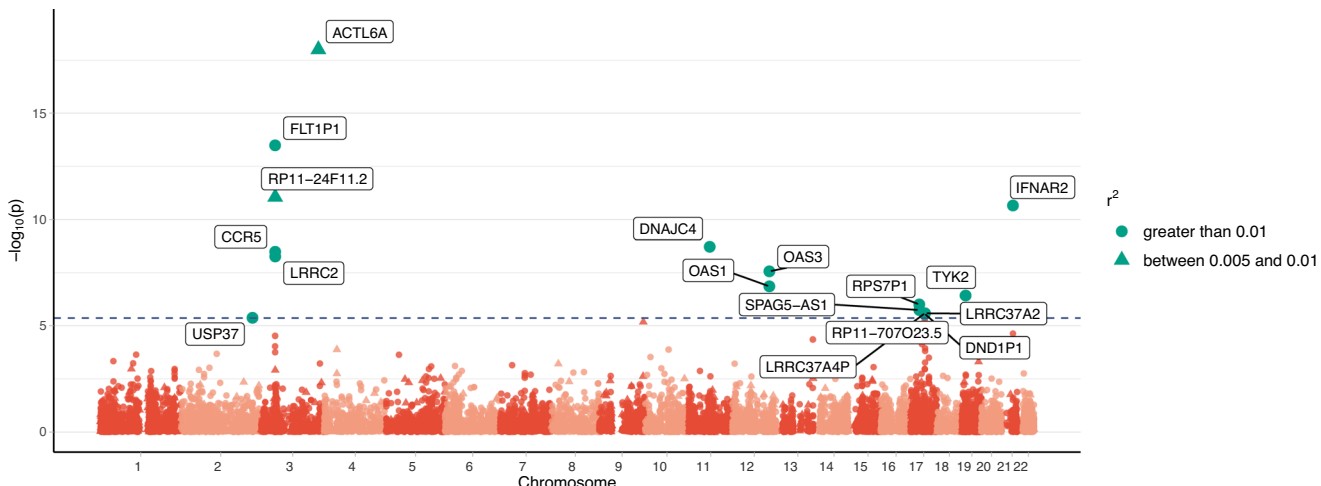

**Fig. 4 | Manhattan plot for COVID-19 severity (B2 outcome) comparing hospitalized COVID-19 patients and controls.** *p*-values were calculated by the SUMMIT (two-sided). The horizontal line marks the genome-wide significance threshold ($0.05/11539 \approx 4.33 \times 10^{-6}$).

As a note, including imputation models with testing $R^2 < 0.01$ increased the burden of multiple tests. To study this, we evaluated SUMMIT's performance for genes with $R^2 \geq 0.01$ under a less stringent *p*-value threshold (as models with $R^2 < 0.01$ were excluded). We confirmed that that the differences in the *p*-value threshold had only a negligible impact on SUMMIT in our real data analyses (Supplementary Fig. 9). SUMMIT identified 3399 gene-trait associations for genes with $R^2 \geq 0.01$ using the less stringent threshold and identified 3339 gene-trait associations for genes with $R^2 \geq 0.01$ when using the more stringent threshold.

### SUMMIT identifies risk genes for COVID-19 severity

We leveraged GWAS summary data from the COVID-19 host genetics initiative (HGI)[25] to identify risk genes for COVID-19 severity. Using SUMMIT, we identified significant associations of 17 genes with COVID-19 severity (B2 outcome) by comparing patients hospitalized with COVID-19 and controls at a Bonferroni-corrected significance threshold of $4.33 \times 10^{-6}$ (Fig. 4). In comparison, the competing methods PrediXcan, TWAS-fusion, UTMOST, and MR-JTI identified 1, 6, 2, and 1 significant genes, respectively (Supplementary Table 1). For the 17 genes identified by SUMMIT, 11 were prioritized by the fine-mapping method FOGS (Table 1). We further validated these 11 genes by analyzing COVID-19 by comparing very severe confirmed respiratory COVID-19 versus population controls (A2). Of them, 10 were validated at $p < 0.05$.

For some of these 11 putative causal genes related to COVID-19 severity, there is already prior knowledge supporting their potential links with COVID-19. To elaborate, SNP *rs1015164*, which lies near the antisense transcribed sequence *RP11-24F11.2*, has been associated with HIV set-point viral load[26,27] and CD4+ T-cell counts. Such chemokine receptor-ligand interactions mediating the traffic of inflammatory cells and pathogen-associated immune responses could plausibly be related to COVID-19 severity. For *FLT1P1*, its expression has been reported to be positively associated with predicted neutrophil count[28]. This may mediate the genetic link between this gene and COVID-19 severity. Another identified gene, *CCR5*, is known to play a role in immune cell migration and inflammation. A study found that *CCR5* blockade in critical COVID-19 patients induced decreased inflammatory cytokines, increased CD8 T cells, and decreased SARS-CoV-2 RNA in plasma[29]. For *OAS1*, both predicted and measured protein levels are inversely associated with COVID-19 susceptibility and severity, which is consistent with the current study's findings[30]. Two of the other genes, namely, *OAS3* and *IFNAR2*, were identified in

our earlier work of COVID-19 TWASs using complementary methods and designs[31].

## Discussion

By leveraging the summary-level expression reference panel with a much larger sample size, our method SUMMIT improved the prediction accuracy of built expression prediction models, which in turn increased the power of identifying risk genes for complex traits.

Through simulations and analyses of the GWAS results for 24 traits, we demonstrated the performance gain of SUMMIT over existing methods. Briefly, we demonstrated that SUMMIT improved the expression imputation accuracy (built more expression prediction models with $R^2 \geq 0.01$), identified more associations, and achieved higher power in identifying "silver standard" genes. Importantly, SUMMIT was applicable in analyzing genes with low expression heritability ($R^2$ between 0.005 and 0.01), which have larger causal effect sizes on complex traits[1] but have not been well captured by existing methods.

SUMMIT can be viewed as a type of gene-based Mendelian randomization (MR) and can provide valid causal interpretations when all genetic variants used in the expression prediction models (with non-zero weights) are valid instrumental variables[32–34]. However, with the widespread horizontal pleiotropy of genetic variables[35], valid instrumental variable assumptions may be violated, and thus, we recommend that practitioners use multiple complementary methods jointly to identify likely causal genes. For example, we can apply fine-mapping approaches such as FOCUS[36] and FOGS[37] to further prioritize likely causal genes by modeling the linkage disequilibrium and correlation among TWAS signals. In addition to fine mapping, it can be useful to complement the TWAS/MR-type approaches with colocalization (in the sense of[38]), which aims to identify causal genetic variants for both gene expression and complex traits. Notably, the existence of colocalized genetic variants (especially those in the cis-acting region) implies that the same variants are responsible for variations in both expression and complex traits, indicating that a causal link between expression and complex traits may exist.

Both SUMMIT and Lassosum[39] are motivated by the recent progress in the estimation of polygenic risk scores using summary-level GWAS data[40,41]. As a result, both Lassosum and SUMMIT construct the primary loss function using penalized regression. However, Lassosum and SUMMIT are different, and SUMMIT is tailored to eQTL summary statistics in the following respects. First, SUMMIT adds an additional step to estimate the LD matrix by utilizing genetic distance

**Table 1 | Predicted gene expression in blood-COVID-19 associations for likely causal genes based on COVID-19 Host Genetics Initiative data**

| Chromosome | Gene | $R^2$ | COVID-B2 | | COVID-A2 | |
|---|---|---|---|---|---|---|
| | | | Direction | p | Direction | p |
| 3 | ACTL6A | 0.008 | – | $9.9 \times 10^{-19}$ | – | $2.5 \times 10^{-1}$ |
| 3 | LRRC2 | 0.044 | – | $5.4 \times 10^{-9}$ | – | $3.9 \times 10^{-5}$ |
| 3 | RP11-24F11.2 | 0.006 | – | $8.8 \times 10^{-12}$ | – | $9.0 \times 10^{-6}$ |
| 3 | FLT1P1 | 0.116 | – | $3.3 \times 10^{-14}$ | – | $1.0 \times 10^{-7}$ |
| 3 | CCR5 | 0.049 | – | $9.9 \times 10^{-19}$ | – | $1.8 \times 10^{-6}$ |
| 12 | OAS1 | 0.056 | – | $1.4 \times 10^{-7}$ | – | $1.2 \times 10^{-8}$ |
| 12 | OAS3 | 0.041 | + | $2.7 \times 10^{-8}$ | + | $3.4 \times 10^{-11}$ |
| 17 | LRRC37A4P | 0.668 | + | $2.6 \times 10^{-6}$ | + | $3.5 \times 10^{-3}$ |
| 17 | RP11-707O23.5 | 0.599 | – | $2.8 \times 10^{-6}$ | – | $3.4 \times 10^{-3}$ |
| 17 | DND1P1 | 0.489 | – | $2.7 \times 10^{-6}$ | – | $3.3 \times 10^{-3}$ |
| 21 | IFNAR2 | 0.037 | – | $2.2 \times 10^{-11}$ | – | $2.2 \times 10^{-9}$ |

"+" and "–" represent positive and negative directions, respectively. p-values were calculated by the SUMMIT (two-sided)

information. Second, Lassosum uses only the LASSO penalty, while SUMMIT considers five different types of penalties. As a result, we have confirmed that SUMMIT achieves much better performance in terms of prediction accuracy and subsequent statistical power in both simulations (Fig. 2) and real data analyses (Fig. 3). Additionally, SUMMIT shares similarities with CoMM-S4[7] as they both use summary-level eQTL data to identify gene-trait associations.

There are several limitations of the current study. First, the summary data of eQTLGen are for whole blood of subjects of European ancestry; thus, the built gene expression prediction models would be applicable only to blood tissue of European ancestry subjects. While SUMMIT can be applied equally to other tissues and ancestry, the corresponding summary eQTL data would be needed for such extensions. Second, several TWAS methods such as UTMOST[11] and MR-JTI[13] have been proposed to leverage expressions from other tissues or functional annotations to improve the prediction accuracy of expression prediction models. Functional annotation databases such as FAVOR[42] may also provide prior information to downweight SNPs that may not contribute to gene expression. We expect that the number of analyzable genes could be increased further if we leveraged information from either other tissues or functional annotations. Third, similar to most existing TWAS methods, the results of SUMMIT imply causality only when valid instrumental variable assumptions are satisfied. A partial solution is to apply fine-mapping to prioritize likely causal genes. However, the robustness of SUMMIT would be significantly improved if we could relax these stringent valid instrumental variable assumptions. We leave this exciting topic for future research.

SUMMIT[43] integrates summary-level eQTL data with GWAS summary statistics via advanced statistical methods. When combined with fine-mapping and functional validations, its findings may yield insights into the genetic basis of diseases and benefit the development of new therapeutic strategies.

## Methods

### Penalized regression model for expression prediction

Consider the following linear regression model for estimating the genetically regulated components of gene expression:

$$\mathbf{Y} = \sum_{j=1}^{p} w_j \mathbf{X}_j + \boldsymbol{\epsilon}, \tag{1}$$

where $\mathbf{Y}$ is the $N$-dimensional vector of gene expression levels of a gene of interest (corrected for important covariates such as age, sex, and principal components of genotypes), $\mathbf{X} = (\mathbf{X}', \cdots, \mathbf{X}')'$ is the $N \times p$

standardized genotype matrix of $p$ cis-SNPs around the gene (within 1 MB of the gene transcription start site and end site), the $p$-dimensional vector $\mathbf{w} = (w_1, \cdots, w_p)'$ is the cis-eQTL effect size, and $\boldsymbol{\epsilon}$ is random noise with a mean of zero.

We estimate $\mathbf{w}$ using a penalized regression framework. Specifically, the objective function is

$$f(\mathbf{w}) = \frac{(\mathbf{Y} - \mathbf{Xw})'(\mathbf{Y} - \mathbf{Xw})}{N} + J_\lambda(\mathbf{w}) = \frac{\mathbf{Y}'\mathbf{Y}}{N} + \mathbf{w}'\left(\frac{\mathbf{X}'\mathbf{X}}{N}\right)\mathbf{w}$$
$$- 2\mathbf{w}'\frac{\mathbf{X}'\mathbf{Y}}{N} + J_\lambda(\mathbf{w}), \tag{2}$$

where $J_\lambda(\cdot)$ is a penalty term. Since the performance of different penalties may vary under different genetic architectures, we consider several penalties, including LASSO[44], elastic net[45], the minimax concave penalty (MCP)[46], the smoothly clipped absolute deviation (SCAD)[47], and MNet[48]. Note that the objective function (Equation (2)) is a function of the marginal statistics $\mathbf{X}'\mathbf{Y}/N$ and the linkage disequilibrium (LD) matrix $\mathbf{X}'\mathbf{X}/N$, and does not require the individual-level data to be observed and stored. This allows us to build expression prediction models using eQTL summary-level data, which are computed using a much larger sample size. That is, we rewrite the objective function as

$$f(\mathbf{w}) = \frac{\mathbf{Y}'\mathbf{Y}}{N} + \mathbf{w}'\mathbf{R}\mathbf{w} - 2\mathbf{w}'\mathbf{r} + J_\lambda(\mathbf{w}), \tag{3}$$

where $\mathbf{r} = \mathbf{X}'\mathbf{Y}/N = (r_1, \cdots, r_p)'$ is a $p$-dimensional vector of standardized marginal effect size for cis-SNPs (i.e., correlation between cis-SNPs and gene expression levels), and $\mathbf{R} = \mathbf{X}'\mathbf{X}/N$ is the LD matrix of the cis-SNPs. We use the z-scores provided in the summary-level eQTL dataset to estimate $\mathbf{r}$ (denoted by $\tilde{\mathbf{r}}$) and use a shrinkage estimator (illustrated below) with an LD reference panel (such as that of the 1000 Genomes Project[49]) to estimate $\mathbf{R}$ (denoted by $\tilde{\mathbf{R}}$). We add an $L_2$ penalty term $\theta\mathbf{w}'\mathbf{w}$ (where $\theta \geq 0$) to the objective function, which ensures a unique solution upon optimization. Note that $\mathbf{Y}'\mathbf{Y}/N$ does not depend on $\mathbf{w}$ and can be ignored when optimizing $f$. Thus, the final objective function that we optimize can be written as,

$$\tilde{f}(\mathbf{w}) = \mathbf{w}'\tilde{\mathbf{R}}\mathbf{w} - 2\mathbf{w}'\tilde{\mathbf{r}} + \theta\mathbf{w}'\mathbf{w} + J_\lambda(\mathbf{w}). \tag{4}$$

The estimates $\hat{\mathbf{w}}$ can be obtained by the coordinate descent algorithm[50], which solves the univariate penalized regression problem sequentially and iteratively. Briefly, suppose that $(\hat{w}_1^{(t)}, \ldots, \hat{w}_p^{(t)})$ are the coefficients in the $t$-th iteration of the coordinate descent algorithm.

Define $z_j^{(t)} = \tilde{r}_j - \sum_{l \neq j} \tilde{R}_{jl} \hat{w}_l^{(t)}$. When $J_\lambda(\mathbf{w})$ is the LASSO penalty ($J_\lambda(\mathbf{w}) = \sum_{j=1}^{p} \lambda |w_j|$), we can update $w_j$ as

$$\hat{w}_j^{(t+1)} = \begin{cases} \frac{z_j^{(t)} - \lambda}{1+\theta} & z_j^{(t)} > \lambda \\ \frac{z_j^{(t)} + \lambda}{1+\theta} & z_j^{(t)} < -\lambda \\ 0 & \text{otherwise} \end{cases} \quad (5)$$

for $j = 1, ..., p$ and $t = 0, 1, ...$.

The convergence properties of the coordinate descent algorithm guarantee a local minimum for $\hat{\mathbf{w}}$[50]. We give the details of the optimization, including the choices of the initial starting values, $\lambda$, and $\theta$, and the updating formulas for the other penalties, in the Supplementary Note 1.

### Estimating the standardized marginal effect size $\tilde{r}$ and LD matrix $\tilde{R}$

The standardized marginal effect size $r_j$ is often not provided in the eQTL summary-level data, but it can be approximated well by $\tilde{r}_j = Z_j / \sqrt{N_j - 1 + Z_j^2}$, where $Z_j$ and $N_j$ are the z-score and sample size for cis-SNP $j$, respectively. The eQTL summary-level data combine the results from multiple cohorts and thus the sample size for each SNP may vary. To obtain an unbiased estimation, we use the SNP-specific sample size $N_j$ instead of the largest sample size (cohort size)[51].

The objective function (4) involves an estimated LD correlation matrix $\tilde{R}$. Instead of using the sample correlation matrix estimated from a reference panel such as 1000 Genomes Project[49] data, we use the shrinkage estimator of the LD matrix[52–54], which stabilizes the results by shrinking the off-diagonal entries toward zero. Specifically, we first calculate the sample LD correlation matrix from a reference panel. Each entry in the LD correlation matrix is then multiplied by the factor $\exp(-\frac{2N_e c_{ij}}{m})$, where $N_e$ is the effective population size, $m$ is the sample size of the data for generating the genetic map, and $c_{ij}$ is the genetic distance between sites $i$ and $j$ in centimorgans. The entries are set to zero if the factor $\exp(-\frac{2N_e c_{ij}}{m})$ is less than a pre-specified threshold $c$. Following others[52,53], we use the genetic distance generated from 1000 Genomes OMNI arrays with $N_e = 11,400$ and $m = 183$ and the prespecified threshold $c$ is set to $1 \times 10^{-3}$.

### Model training and evaluation

We trained our expression prediction models by using the cis-eQTL summary-level data from eQTLGen[6], which consist of effect sizes of >11 million SNPs from 31,684 blood samples. Following PrediXcan[8], SNPs in the vicinity of the given gene (within 1 Mbp of the gene transcription start site and end site) were used as the cis-genotype information. Furthermore, we filtered out all SNPs with minor allele frequency (MAF) < 0.01 and those that were nonbiallelic, ambiguous or not included in the HapMap3 SNP set[8].

We used both genotype and gene expression data from the GTEx project (version V7, dbGaP Accession number phs000424.v7.p2, https://www.gtexportal.org/home/datasets)[55] to select the tuning parameters. The processed gene expression values in whole blood ($N = 369$) were downloaded from the GTEx website. Briefly, the RPKMs in each sample were standardized and normalized by quantile transformation. The expression for each gene was further adjusted for sex, genotyping platform, 35 PEER factors and three genotype-based principal components (PCs) and the residuals were used as the processed expression levels. We used the squared correlation between the predicted and observed expressions (that is, $R^2$) to select the best tuning parameters. Notably, the subjects in GTEx v6 ($N = 336$; 1.1%) were meta-analyzed in eQTLGen[6] and may result in suboptimal tuning parameters.

We used independent data of subjects who were included in GTEx v8 but not in GTEx v7 ($N = 309$) for external validation. Notably, the subjects in GTEx v8 were not meta-analyzed in eQTLGen and thus can be viewed as an independent dataset for external validation. Because genes with low expression heritability have substantially larger causal effect sizes on complex traits[1], we selected models with $R^2 \geq 0.005$ instead of the commonly used criterion of $R^2 \geq 0.01$. The threshold ($R^2 \geq 0.005$) was justified by an informal theoretical investigation using a well-established statistical theory by Cramer[56]. Briefly, assuming a standard multiple regression model, Cramer[56] showed that under the null hypothesis of $\beta = 0$, $R^2$ follows a beta distribution, i.e., $R^2 \sim \mathcal{B}((p-1)/2, (n-p)/2)$. In SUMMIT, we used the eQTL-gen summary-level data with $n = 31,684$ and the median number of SNPs with nonzero weights for each gene was $p = 34$, leading to $R^2 \sim \mathcal{B}(16.5, 15825)$ under the null hypothesis. The rejection region $\approx (0.00263, 1]$ (under the transcriptome-wide significance level of $\alpha = 0.05/16884 \simeq 3.0 \times 10^{-6}$). The above derivation, however, ignores the impact of regularization induced by penalized regression. To consider the potential impact of regularization, we propose using a slightly conservative threshold of $R^2 \geq 0.005$ for SUMMIT. As a note, formally considering the regularization bias is nontrivial and requires additional assumptions; and we leave such interesting topics for future research.

### Association analyses with individual expression prediction models

When individual-level GWAS data (genotype data $\mathbf{X}_{\text{new}}$, phenotype $\mathbf{P}_{\text{new}}$, and covariance matrix $\mathbf{C}_{\text{new}}$) are available, one can apply a generalized linear regression model

$$f(E[\mathbf{P}_{\text{new}} | \mathbf{X}_{\text{new}}, \mathbf{C}_{\text{new}}]) = \alpha \mathbf{C}_{\text{new}} + \beta \mathbf{X}_{\text{new}} \hat{\mathbf{w}} \quad (6)$$

to test $H_0 : \beta = 0$, where $f(\cdot)$ is a link function, and $\mathbf{X}_{\text{new}} \hat{\mathbf{w}}$ is the predicted genetically regulated expression for the trait of interest.

When only summary-level GWAS data are available, one can apply a burden-type test:

$$\tilde{Z} = \mathbf{Z} \hat{\mathbf{w}} / \sqrt{\hat{\mathbf{w}}' \mathbf{V} \hat{\mathbf{w}}}, \quad (7)$$

where $\mathbf{Z}$ is the vector of z-scores for all cis-SNPs and $\mathbf{V}$ is the LD matrix of analyzed SNPs (which can be estimated by using a population reference panel such as that of the 1000 Genomes Project[49]).

### Association analyses with multiple expression prediction models

To further improve the power, we apply the Cauchy combination test[18] to integrate information from $K$ models with $R^2 \geq 0.005$. Specifically, we use the following test statistics:

$$T = \sum_{j=1}^{K} \tilde{R}_j^2 \tan\{(0.5 - p_j)\pi\}, \quad (8)$$

where $p_j$ is the p-value for model $j$ and $\tilde{R}_j^2$ is calculated by $R_j^2 / \sum_{j=1}^{k} R_j^2$. $T$ approximately follows a standard Cauchy distribution, and the p-value can be calculated as $0.5 - \arctan(T)/\pi$. Notably, we use $\tilde{R}_j^2$ as the weights when combining multiple expression prediction models because a larger $\tilde{R}_j^2$ indicates a better expression prediction model. The Cauchy combination test has been widely used in the human genetics community[18,57], because the p-value approximation is accurate for highly significant results (which are of interest) and there is no need to estimate the correlation structure among the combined p-values.

One may be interested in the association direction for a specific gene of interest. For a majority of the significant genes identified by SUMMIT, all the expression prediction models yield the same association direction. When the expression prediction models provide conflicting association directions, we determine the association direction by majority voting. In the rare situation in which the number

of models indicating positive associations is the same as the number of models indicating negative associations, we declare the association direction unknown.

## Simulation study design

We conducted simulation studies to evaluate how the sample size of the expression reference panel impacts the expression prediction accuracy and the subsequent power of TWASs. Additionally, we evaluated whether using the summary-level eQTL data yielded similar performance to that of using the individual-level expression reference panel. Specifically, we used data from the UK Biobank and randomly chose genotype data from 31,684 (to match the sample size of the eQTLGen data) unrelated white British individuals as training data, genotype data from an additional 369 (to match the sample size of the GTEx v7 data) unrelated white British individuals as tuning data, and genotype data from an additional 10,000 unrelated white British individuals as test data. The imputed data of 877 *cis*-SNPs (with MAF > 1%, Hardy-Weinberg *p*-value > $10^{-6}$, and imputation "info" score > 0.4) of the arbitrarily chosen gene *CHURC1* were used for our main simulations. We also considered several other randomly selected genes (Supplementary Figs. 3–6).

We simulated gene expression levels and phenotype values by $\mathbf{E}_g = \mathbf{X}\mathbf{w} + \epsilon_e$ and $\mathbf{Y} = \beta \mathbf{E}_g + \epsilon_p$, respectively. $\mathbf{X}$ is the standardized genotype matrix, $\mathbf{w}$ is the effect size, the scalar $\beta$ is the association coefficient, $\epsilon_e \sim N(0, 1 - h_e^2)$, and $\epsilon_p \sim N(0, 1 - h_p^2)$, where $h_e^2$ and $h_p^2$ are the expression heritability (i.e., the proportion of gene expression variance explained by SNPs) and phenotypic heritability (i.e., the proportion of phenotypic variance explained by gene expression levels), respectively. We randomly selected $p_{causal}$, that is, the proportion of SNPs that are causal, and generated its effect size $w_j$ from $N(0, 1)$. The effect sizes for the remaining noncausal SNPs were set to 0. We rescaled the effect sizes $w$ and $\beta$ to achieve the targeted $h_e^2$ and $h_p^2$.

To evaluate the performance of the proposed SUMMIT method, we performed an association scan on the whole simulated training data ($\mathbf{E}_g$, $\mathbf{X}$) and computed the summary-level data (i.e., z-scores) using a linear regression. To study the impact of the sample size of the training data, we also built prediction models using training data of different sample sizes (300, 600, 3000, 10,000, 31,684). We compared SUMMIT with two widely used methods, PrediXcan[8] and TWAS-fusion[9]. Furthermore, we investigated the idea of using a polygenic risk score method (e.g., Lassosum[39]) to train the expression prediction models. We trained models with PrediXcan and TWAS-fusion using individual-level data of 670 samples (to match the sample size of blood tissue in the GTEx v8 data). As a note, in addition to Lassosum, we only compared SUMMIT with PrediXcan and TWAS-fusion in simulations because all of these methods focus on single-tissue information. While leveraging cross-tissue information can further improve the performance as demonstrated in UTMOST[11] and MR-JTI[13], it is not our focus here, and thus, we did not compare cross-tissue methods such as UTMOST and MR-JTI in our simulations, leaving such interesting topics for future research.

We considered comprehensive scenarios that varied the proportion of causal SNPs $p_{causal}$ (0.01, 0.05, 0.1, 0.2), expression heritability $h_e^2$ (0.005, 0.01, 0.1), and phenotypic heritability $h_p^2$ (0.1, 0.2, 0.5, 0.8). For each scenario, we repeated the simulations 1000 times. The statistical power was calculated as the proportion of 1000 repeated simulations with a *p*-value less than the genome-wide significance threshold $0.05/20,000 = 2.5 \times 10^{-6}$.

## Comparison with existing methods

We further compared SUMMIT with several TWAS methods, including Lassosum[39], MR-JTI[13], PrediXcan[8], TWAS-fusion[9], and UTMOST[11], for whole blood tissue in the following respects. Lassosum is a polygenic risk score method that can be used to build expression prediction models with a summary-level reference panel. After building the

expression prediction models, we apply the standard TWAS framework to obtain the results. PrediXcan uses Elastic Net to build gene expression prediction models; TWAS-fusion applies several methods, including BLUP, BSLMM, Elastic Net, LASSO, and TOP1 to build expression prediction models. MR-JTI and UTMOST leverage cross-tissue information when building gene expression prediction models. All four TWAS methods are based on an individual-level expression reference panel, while our method SUMMIT and Lassosum are based on a summary-level expression reference panel.

First, we compared the prediction accuracy (in terms of $R^2$) estimated by different methods. Notably, while the prediction performances of the models developed using competing methods were estimated through cross validation, the prediction performances of the models developed using SUMMIT and Lassosum were estimated in an external testing dataset. This difference may slightly favor PrediXcan and TWAS-fusion. The difference in $R^2$ across genes was tested by the one-sided Kolmogorov-Smirnov test, a nonparametric test that calculates the largest distance between the empirical distribution functions to determine whether two distributions are equivalent.

Second, we compared different methods by analyzing GWAS summary statistics for 24 complex traits. The details of the 24 traits are summarized in Supplementary Data 1. We used the Bonferroni correction for each method with different significance thresholds as different methods have different numbers of analyzable genes. To make a fair comparison, we also evaluated a common gene set that can be analyzed by all methods and used the same Bonferroni-corrected significance threshold to determine the significant gene sets. The numbers of significant genes identified by the different methods were further compared by the Wilcoxon signed-rank test, which compares two matched samples to test whether their population mean ranks differ.

Third, as a TWAS can be viewed as a special case of Mendelian randomization[58], we further compared different methods in terms of identifying the causal genes that mediate the associations between GWAS loci and the traits of interest. Following Barbeira et al.[20], we curated a set of likely causal gene-trait pairs using information that was independent of the GWAS results. Briefly, we utilized the OMIM database[21] and rare variant results from exome-wide association studies[22–24], obtaining 1, 287 gene-trait pairs. We used LDetect to partition the genome into approximately independent LD blocks[59] and refined the gene-trait pairs by considering only the genes that were located in LD blocks with at least one genome-wide significant variant, leading to 148 likely causal gene-trait pairs (among 24 distinct traits). We compared different methods by the area under the receiver operating characteristic curve (AUC).

## Applications to COVID-19 GWAS data

To identify genes associated with COVID-19 severity, we applied SUMMIT-derived models to GWAS summary data from the COVID-19 HGI (Release 5 (January 2021))[25]. The detailed information of participating studies, quality control, and analyses are included on the COVID-19 HGI website (https://www.covid19hg.org/results/). Briefly, data from 9, 986 hospitalized COVID-19 patients and 1, 877, 672 population controls were used in the current analyses. Hospitalized COVID-19 cases included patients who (1) had laboratory confirmed SARS-CoV-2 infection (RNA- and/or serology-based) and (2) were hospitalized due to corona-related symptoms. The controls are subjects who are not cases. Only individuals of European ancestry were included to ensure a homogeneous population structure for the analyses. A fixed-effect meta-analysis of the individual participating studies was performed and variants with imputation quality > 0.6 were retained.

We applied the fine-mapping method FOGS[37] to prioritize likely causal genes for COVID-19 severity. We evaluated the associations of the identified genes with an additional COVID-19 phenotype. Briefly,

we leveraged A2_ALL_eur (Europeans; 5, 101 cases and 1, 383, 241 controls) to compare very severe confirmed respiratory COVID-19 vs. population controls.

## Reporting summary

Further information on research design is available in the Nature Research Reporting Summary linked to this article.

## Data availability

The GWAS summary data used in this study are summarized in Supplementary Data 1 (with the download link). The eQTL summary data are available at https://www.eqtlgen.org/cis-eqtls.html. The COVID-19 HGI summary data can be downloaded from https://www.covid19hg.org/results/. The UK Biobank is an open-access resource but requires registration, available at https://www.ukbiobank.ac.uk/researchers/. The genotype and RNA sequencing data for the GTEx project are available at the database of Genotypes and Phenotypes (accession number phs000424.v8.p2, https://www.ncbi.nlm.nih.gov/projects/gap/cgi-bin/study.cgi?study_id=phs000424.v8.p2). The processed gene expression for the GTEx project is available from the GTEx portal (https://gtexportal.org). The MR-JTI, PrediXcan, and UTMOST models can be downloaded from https://doi.org/10.5281/zenodo.3842289. The TWAS-fusion's model can be downloaded from http://gusevlab.org/projects/fusion/. The 1000 Genomes Project data can be downloaded from https://www.internationalgenome.org/data. The genetic distance data for 1000 Genomes Project can be downloaded from https://github.com/joepickrell/1000-genomes-genetic-maps. The SUMMIT models generated in this study are available from OSF.IO at https://doi.org/10.17605/OSF.IO/7MXSA. The raw data and code to replicate figures and tables in the manuscript are available from OSF.IO at https://doi.org/10.17605/OSF.IO/FJPDU. All real data results are available at https://chongwulab.shinyapps.io/SUMMIT-app/, where practitioners can search and download results easily. All other data are available in the paper and its supplementary information files. Source data are provided with this paper.

## Code availability

The SUMMIT software is available on GitHub (https://github.com/ChongWuLab/SUMMIT) and Zenodo[43]. The codes and corresponding data for reproducing the results described in this study are available on OSF.IO[60].

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

## Acknowledgements

National Institutes of Health (R03 AG070669) supported Z.Z., J.R.B., and C.W. This study was conducted using the UK Biobank recourse under Application Number 48240 (https://www.ukbiobank.ac.uk/researchers/). The content is solely the responsibility of the authors and does not necessarily represent the official views of the National Institutes of Health. The Genotype-Tissue Expression (GTEx) Project was supported by the Common Fund of the Office of the Director of the National Institutes of Health, and by NCI, NHGRI, NHLBI, NIDA, NIMH, and NINDS. The authors would like to thank all of the individuals for their participation in the GWASs and UK Biobank and all the researchers, clinicians, technicians and administrative staff for their contribution to the studies and for making their GWAS summary results publicly available.

## Author contributions

C.W. conceived and designed the study. Z.Z. and C.W. developed the computational algorithms and wrote the SUMMIT program. Z.Z. performed the real data analysis and simulations. Z.Z. created the website that curated the results. Y.B. tested the program and drew the workflow diagram of SUMMIT. J.R.B. and L.W. provided critical feedback and contributed to the interpretation of the results. All authors wrote and proofread the manuscript.

## Competing interests

The authors declare no competing interests.
