## [Peer Review File · Nature Communications]

SUMMIT: An integrative approach for better transcriptomic data imputation improves causal gene identificationREVIEWER COMMENTS

Reviewer #1 (Remarks to the Author):

Zhang et al. present a generally well organized and clearly presented manuscript focused on the idea that leveraging larger gene expression eQTL reference data can improve the power and overall utility of approaches that systematically integrate GWAS with eQTL data (such as PrediXcan and TWAS/FUSION). To test the idea, the authors present a new method called SUMMIT which allows them to derive improved gene expression prediction models using eQTL summary statistics, thus allowing them to leverage larger sample sizes from consortia such as eQTLGen. The authors test and demonstrate their methods through simulation studies, evaluation of gene expression prediction in GTEx, and application to interpretation for a real GWAS of COVID-19. The findings are conceptually interesting and technically well developed. Overall, I have several points that require clarification, pertaining primarily to the need for (a) a clearer description of which methods are being compared and why, and (b) more direct comparison of results across methods by benchmarking using a single data set across methods (e.g. GTEx) rather than only comparing PrediXcan and TWAS from GTEx versus SUMMIT using eQTLGen (which is not a direct comparison). My detailed comments are provided below.

Major:

1. Introduction: Where you talk about the number of “analyzable” and its effect on power. Please clarify what factors determine whether a gene is analyzable, as some readers may not be familiar with the concept. Also, in your example noting an increase in analyzable genes when the “reference panel increased from 338 samples to 369 samples” – please provide a literature citation for this statement. Also, clarify whether sample size was the only thing that changed, or if other factors could have yielded the increase in analyzable genes.
2. Results - overview (p. 4): Please provide a rationale for using the Cauchy combination test, as readers may not be familiar with it.
3. Results - simulation studies and elsewhere: In the simulation studies (p. 4), you state a comparison in prediction performance for TWAS, PrediXcan and SUMMIT. Later, in the real data analyses (p. 7), you add comparison to MR-JTL and UTMOST. Please provide some context by indicating what is the actual methodological difference between the methods under comparison. In addition, please clarify why some comparisons include MR-JTL and UTMOST, and others do not.
4. Overall (related to point above): Throughout the manuscript results and elsewhere, you make claims about the prediction performance of TWAS vs. PrediXcan vs. SUMMIT. Sometimes you attribute these differences to sample size of the corresponding reference panels. Thus, it can be confusing to interpret the differences in performance across methods. Throughout the Results, I suggest adding a direct comparison of SUMMIT using reference panels (or simulated data sets) or different sizes, which would allow you to compare the actual method performance across different scenarios.
5. Figure 2b (statistical power): I believe this plot shows power of the association test examining the relationship between predicted expression and phenotypic outcome. Please clarify what model was used to simulate the relationship between predicted expression and the phenotype of interest.
6. Discussion: You note “the results of SUMMIT imply causality only when valid instrumental variable assumptions are met”. It would be worth adding to the paragraph that, in addition to fine mapping, it can be useful to complement the TWAS / MR type approaches with colocalization (in the sense of Giambartolomei 2014).
7. Overall: Throughout the manuscript, you mention multiple times that you want to relax the cutoff on proportion of heritability explained in expression ($h^2 \geq 0.005$ rather than the typical cutoff of $h^2 \geq 0.01$). Both cutoffs seem arbitrary and it would help the manuscript if you could derive a reasonable cutoff on heritability h^2 as a function of sample size (similar to how people examine GWAS SNPs based on a cutoff of observed minor allele count [MAC] which is a function of sample size, rather than minor allele frequency [MAF] which is arbitrary).
8. Overall: The idea of using large-scale eQTL summary statistics to improve expression prediction is useful. Summary statistics based polygenic risk scores have already been implemented (e.g. Lassosum method in Mak 2017). Please clarify the difference between SUMMIT versus use of

lassosum to develop predictors of expression, and consider including a direct comparison in the manuscript.

Minor:

1. Some awkward wording. See for example:

a. Introduction: "First, the expression reference panel..." – "the" should be "an".

b. Introduction: "...identifies way more risk genes" is too informal.

c. Discussion: "we can apply fine-mapping approaches such FOCUS"- need to add "as" before "FOCUS".

d. Discussion: "informative information" is an awkward phrase.

2. Figure 4: Please clarify in the legend that the Manhattan plot shows association results based on analysis using SUMMIT (with eQTLGen as the reference panel?). This point seems clear based on the text, but it will be better to make the legend clear.

Reviewer #2 (Remarks to the Author):

Zhang et al presented a new method to impute gene expression for TWAS. The method is novel that it can utilize summary-level eQTL data to build gene expression models, while most current methods need individual-level genotype data as input. This feature significantly improves prediction power for gene expression in blood by using the eQTLgen dataset. It also increased the number of genes with gene prediction model that can be used in TWAS. Application of this method in the COVID GWAS data leads to more significant genes identified compared to before. The method is clearly presented. It would be very useful for the field to have a tool that uses summary level eQTL data for building more powerful gene prediction models for TWAS.

The paper clearly shows that improved performance of prediction models in Figure 2, especially for lower heritability and bigger p_{causal} genes. One thing that is not clear and can be studied by simulation studies is how this translates to improvement for the TWAS power. Even though low h^2 genes have been shown to show have higher causal effect by the MESC paper and SUMMIT provides a better model, these low R^2 genes ($R^2 < 0.01$) may still be hard to detect and increase the burden of multiple testing. I suggest the authors to further investigate how this improvement in expression prediction power affect TWAS power.

The application of the method identified more significant genes. Explanation about why these genes were only identified by SUMMIT would be helpful for the readers to understand where the gain of power comes from. Most significant genes identified by SUMMIT and not by other method has $R^2 > 0.01$, is it because Predixcan or TWAS's model is not as accurate? I think simulations regarding TWAS power for $R^2 > 0.01$ genes may help to answer this.

Another information that should be included in the manuscript is the connection with current method. It seems to me the impute expression method is similar/same to the "lasso sum" method developed for calculating PRS (Timothy Shin Heng Mak et al, 2017 Genetic Epidemiology), except that the method used here has been further extended it to use other penalties in addition to the lasso penalty. I think this point should be made clear in the manuscript and give proper credit to the lasso sum method. Another point is that this is not the first time that a summary level eQTL data be used for TWAS, a recent method has been published (Yang, Yi et al. "CoMM-S4: A Collaborative Mixed Model Using Summary-Level eQTL and GWAS Datasets in Transcriptome-Wide Association Studies." *Frontiers in genetics* vol. 12 704538. 20 Sep. 2021, doi:10.3389/fgene.2021.704538"). I think current method has the benefit of easy implementation and integration with popular TWAS framework. But proper reference should be given to previous publications.

Minor point: it is not necessary to have subtitles for the figures, relevant information should be provided in figure legend.

Point-by-point Response to Reviewers' Comments

Response: Reviewer 1 (Previous submission number: NCOMMS-21-46227A)

SUMMIT: An integrative approach for better transcriptomic data imputation improves causal gene identification

Date: July 26, 2022

Comment: *“Zhang et al. present a generally well organized and clearly presented manuscript focused on the idea that leveraging larger gene expression eQTL reference data can improve the power and overall utility of approaches that systematically integrate GWAS with eQTL data (such as PrediXcan and TWAS/FUSION). To test the idea, the authors present a new method called SUMMIT which allows them to derive improved gene expression prediction models using eQTL summary statistics, thus allowing them to leverage larger sample sizes from consortia such as eQTLGen. The authors test and demonstrate their methods through simulation studies, evaluation of gene expression prediction in GTEx, and application to interpretation for a real GWAS of COVID-19. The findings are conceptually interesting and technically well developed. Overall, I have several points that require clarification, pertaining primarily to the need for (a) a clearer description of which methods are being compared and why, and (b) more direct comparison of results across methods by benchmarking using a single data set across methods (e.g. GTEx) rather than only comparison PrediXcan and TWAS from GTEx versus SUMMIT using eQTLGen (which is not a direct comparison). My detailed comments are provided below.”*

Reply: Thank you for your kind support and recognition of our study. We are very grateful for your helpful summary, insightful comments, and constructive suggestions, which led to a significantly improved manuscript! We have made every effort to address the comments you raised. In particular, we have add descriptions of which methods are being compared on Page 21 and explained why we only compared PrediXcan, TWAS, and Lassosum (newly added as suggested) in simulations on Page 21.

As a remark, PrediXcan and TWAS-Fusion are designed for individual-level expression reference panel, while SUMMIT is designed for summary-level expression reference panel (often with a much larger sample size), making it hard to compare them directly. To further highlight the benefits of SUMMIT, we have further compared SUMMIT with Lassosum using

the same training data in both simulations and real data analyses (i.e., eQTLGen). We have also investigated how the sample size of reference expression panels affects the performance of proposed method SUMMIT (Figure 2c on Page 6), highlighting the benefits of using a larger sample to train the expression prediction models.

Finally, we have rerun all simulations with a saved seed to further improve the reproducibility of our studies. The simulation results were changed very slightly, and the conclusions remained the same. When replicating our results, we noticed that some real data results were based on different (and old) GWAS data caused by a server issue. We fixed this issue; the real data results were changed slightly, but the conclusions remained the same. All simulation and real data results have been exactly replicated several times by internal users, and the corresponding tutorials and codes have also been provided.

We have summarized our point-by-point responses below. Your original comments are in italics. We have also highlighted the major changes in blue in the manuscript.

Comment: *“Introduction: Where you talk about the number of “analyzable” and its effect on power. Please clarify what factors determine whether a gene is analyzable, as some readers may not be familiar with the concept. Also, in your example noting an increase in analyzable genes when the “reference panel increased from 338 samples to 369 samples” – please provide a literature citation for this statement. Also, clarify whether sample size was the only thing that changed, or if other factors could have yielded the increase in analyzable genes. ”*

Reply: Thank you for your helpful comments. We have clarified that the analyzable genes are defined as genes with satisfactory gene expression prediction models (e.g., prediction accuracy $R^2 \geq 0.01$). The sample size was the only thing that changed for an increase in analyzable genes when the “reference panel increased from 338 samples to 369 samples” and we have added citations for this. We have also cited another study to further support this argument. [2] show that among the 44 overlapping tissues in GTEx, the average number of analyzable genes increased from 4,570 (v6p) to 7,213 (v8) for one popular TWAS method PrediXcan when the average sample increased from 160 (v6p) to 332 (v8). We have revised the manuscript on Page 2 accordingly.

Comment: *“Results - overview (p. 4): Please provide a rationale for using the Cauchy combination test, as readers may not be familiar with it.”*

Reply: Thank you for your insightful comments. We have added justifications for using the Cauchy combination test on Page 4. Briefly, as p -values from different gene expression prediction models can be correlated, we apply the Cauchy combination test to aggregate p -values from the fitted prediction models, which effectively quantifies the overall gene-trait associations. Cauchy combination test is a computationally efficient p -value combination method that provides accurate p -value approximation for the highly significant results (which are of interest) and does not require estimating the correlation structure among the combined p -values.

Comment: *“Results - simulation studies and elsewhere: In the simulation studies (p. 4), you state a comparison in prediction performance for TWAS, PrediXcan and SUMMIT. Later, in the real data analyses (p. 7), you add comparison to MR-JTL and UTMOST. Please provide some context by indicating what is the actual methodological difference between the methods under comparison. In addition, please clarify why some comparisons include MR-JTL and UTMOST, and others do not.”*

Reply: We thank the reviewer for pointing this out. PrediXcan uses the Elastic Net to build gene expression prediction models; TWAS-fusion applies several methods, including BLUP, BSLMM, Elastic Net, LASSO, and TOP1 to build expression prediction models. Both MR-JTI and UTMOST leverage the cross-tissue information when building gene expression prediction models. All these four methods are based on an individual-level reference panel, while our proposed method SUMMIT is based on a summary-level reference panel. We have added the descriptions on Page 21.

In our simulations, we compared SUMMIT with PrediXcan, TWAS, and Lassosum (newly added in this version) because they all focused on using single-tissue information. While leveraging cross-tissue information can further improve the performance as demonstrated in MR-JTI and UTMOST, it is not our focus here, and thus we did not compare MR-JTI and UTMOST in our simulations and leave such interesting topics to future research (as discussed on Page 14). In our real data analyses, we further compared SUMMIT with MR-JTI and UTMOST because the corresponding gene expression prediction models are publicly available. Furthermore, we are curious about the relative performances among different types of TWAS methods in real data analyses (and expect our readers may also be curious about that). We have clarified this point on Page 20-21.

Comment: *“Overall (related to point above): Throughout the manuscript results and elsewhere, you make claims about the prediction performance of TWAS vs. PrediXcan vs. SUMMIT. Sometimes you attribute these differences to sample size of the corresponding reference panels. Thus, it can be confusing to interpret the differences in performance across methods. Throughout the Results, I suggest adding a direct comparison of SUMMIT using reference panels (or simulated data sets) or different sizes, which would allow you to compare the actual method performance across different scenarios.”*

Reply: Thank you so much for this insightful and extremely helpful suggestion. Following your suggestion, we conducted additional simulations (Figure 2c) and studied the influence of different sample sizes of reference panels on gene expression prediction accuracy and its subsequent statistical power in identifying gene trait associations. We varied the expression reference panel sample size N from (300, 600, 3000, 10000, 31684), expression heritability h_e^2 from (0.005, 0.01, 0.05), and set phenotypic heritability $h_p^2 = 0.2$. We observed that the TWAS power and prediction accuracy were highly correlated. As the sample size of the expression reference panel increased, the expression prediction models became more accurate, leading to higher TWAS power. Of note, due to the setups (i.e., two-sample framework) of the simulations, the gains in the sample size of the expression reference panel can only interact with TWAS power through better prediction models. We have also added a polygenic risk score method Lassosum in both simulations and real data analyses using the same summary-level expression reference panel to train the gene expression prediction models.

Comment: *“Figure 2b (statistical power): I believe this plot shows power of the association test examining the relationship between predicted expression and phenotypic outcome. Please clarify what model was used to simulate the relationship between predicted expression and the phenotype of interest.”*

Reply: Thank you for your careful reading. Yes, you are right. We have clarified this on Page 5. We have also added simulation details on Page 20. Briefly, we simulated gene expression levels and phenotype values by $E_g = Xw + \epsilon_e$ and $Y = \beta E_g + \epsilon_p$, respectively, where X is standardized genotype matrix, w is the effect size, scalar β is the association coefficient of interest, $\epsilon_e \sim N(0, 1 - h_e^2)$, and $\epsilon_p \sim N(0, 1 - h_p^2)$, h_e^2 and h_p^2 were the expression

heritability and phenotypic heritability to be pre-specified. We re-scaled the effect sizes w and β to achieve the targeted h_e^2 and h_p^2 .

Comment: *“Discussion: You note “the results of SUMMIT imply causality only when valid instrumental variable assumptions are met”. It would be worth adding to the paragraph that, in addition to fine mapping, it can be useful to complement the TWAS / MR type approaches with colocalization (in the sense of Giambartolomei 2014).”*

Reply: Thank you for pointing this out. We totally agree that colocalization should be a crucial component of transcriptome-wide association studies. According to your suggestion, we have added discussions on Page 14.

Comment: *“Overall: Throughout the manuscript, you mention multiple times that you want to relax the cutoff on proportion of heritability explained in expression ($h^2 \geq 0.005$ rather than the typical cutoff of $h^2 \geq 0.01$). Both cutoffs seem arbitrary and it would help the manuscript if you could derive a reasonable cutoff on heritability h^2 as a function of sample size (similar to how people examine GWAS SNPs based on a cutoff of observed minor allele count [MAC] which is a function of sample size, rather than minor allele frequency [MAF] which is arbitrary).”*

Reply: Thank you for raising an excellent point. We fully agree that both cutoffs seem arbitrary and have added explanation on how the cutoff of h^2 can be lowered by the increase in sample size n . We were able to find a well-established theoretical-based statistical paper [1] on the asymptotic distribution of the R^2 . Briefly, assuming a standard multiple regression model, [1] showed that under the null hypotheses of $H_0 : R^2 = 0$, R^2 follows a beta distribution, i.e., $R^2 \sim \mathcal{B}((p-1)/2, (n-p)/2)$. In SUMMIT, we used the eQTL-gen summary-level data with $n = 31,684$ and the median number of SNPs with non-zero weights for each gene is $p = 34$, leading to $R^2 \sim \mathcal{B}(16.5, 15825)$ under the null. The rejection region $\approx (0.00263, 1]$ (under transcriptome-wide significance level $\alpha = 0.05/16884 \simeq 3.0 \times 10^{-6}$). In the above derivation, we ignored the impact of regularization induced by penalized regression. To consider the potential impact of regularization, we propose using $R^2 \geq 0.005$ as the cutoff for SUMMIT. As a remark, formally considering regularization bias is not trivial and requires additional assumptions; and we hope to leave such interesting topics to future research. We have added descriptions on Page 18.

Comment: *“Overall: The idea of using large-scale eQTL summary statistics to improve expression prediction is useful. Summary statistics based polygenic risk scores have already been implemented (e.g. Lassosum method in Mak 2017). Please clarify the difference between SUMMIT versus use of Lassosum to develop predictors of expression, and consider including a direct comparison in the manuscript.”*

Reply: Thank you for your helpful and excellent comments. We totally agree that polygenic risk scores methods such as Lassosum can be equally applied to build gene expression prediction models. Particularly, both SUMMIT and Lassosum constructed the primary loss function using penalized regression. However, Lassosum and SUMMIT are different and SUMMIT is tailored to eQTL summary statistics in the following aspects. First, SUMMIT adds an additional step to estimate the LD matrix by utilizing genetic distance information. Second, Lassosum only uses LASSO penalty, while SUMMIT considers five different types of penalties. As a results, we have confirmed that SUMMIT achieves better performance in terms of prediction accuracy and subsequent statistical power in both simulations (Figure 2 on Page 6) and real data analyses (Figure 3 on Page 10). We have added discussions on Page 14.

Comment: *“Minor: 1. Some awkward wording. See for example:*

- a. Introduction: “First, the expression reference panel. . .” – “the” should be “an”.*
- b. Introduction: “. . . identifies way more risk genes” is too informal.*
- c. Discussion: “we can apply fine-mapping approaches such FOCUS”- need to add “as” before “FOCUS”.*
- d. Discussion: “informative information” is an awkward phrase.*

2. Figure 4:

Please clarify in the legend that the Manhattan plot shows association results based on analysis using SUMMIT (with eQTLGen as the reference panel?). This point seems clear based on the text, but it will be better to make the legend clear.”

Reply: Thank you for your careful reading. We have carefully proofread the manuscript several times and made every effort to correct typos and replace awkward wordings.

Thank you for your helpful suggestions. We have added information to Figure 4 as you suggested.

Response: Reviewer 2 (Previous submission number: NCOMMS-21-46227A)

SUMMIT: An integrative approach for better transcriptomic data imputation improves causal gene identification

Date: July 26, 2022

Comment: *“Zhang et al presented a new method to impute gene expression for TWAS. The method is novel that it can utilize summary-level eQTL data to build gene expression models, while most current methods need individual-level genotype data as input. This feature significantly improves prediction power for gene expression in blood by using the eQTLgen dataset. It also increased the number of genes with gene prediction model that can be used in TWAS. Application of this method in the COVID GWAS data leads to more significant genes identified compared to before. The method is clearly presented. It would be very useful for the field to have a tool that uses summary level eQTL data for building more powerful gene prediction models for TWAS.”*

Reply: Thank you for your very kind support and recognition of our method. We are grateful for your constructive and insightful comments and suggestions, which led to a significantly improved manuscript! We have made every effort to address the comments you raised.

As a remark, we have rerun all simulations with a saved seed to further improve the reproducibility of our studies. We fixed a server issue in real data analyses; the results were changed slightly, but the conclusions remained the same. All simulation and real data results have been exactly replicated several times by internal users, and the corresponding tutorials and codes have also been provided.

We have summarized our point-by-point responses below. Your original comments are in italics. We have also highlighted the major changes in blue in the manuscript.

Comment: *“The paper clearly shows that improved performance of prediction models in Figure 2, especially for lower heritability and bigger p_{causal} genes. One thing that is not clear and can be studied by simulation studies is how this translates to improvement for the TWAS power. Even though low h^2 genes have been shown to show have higher causal effect by the MESC paper and SUMMIT provides a better model, these low R^2 genes ($R^2 < 0.01$) may still be hard to detect and increase the burden of multiple testing. I suggest the authors*

to further investigate how this improvement in expression prediction power affect TWAS power.”

Reply: We appreciate your insightful and helpful comments. Following your suggestions, we have conducted simulation studies targeting genes with estimated $R^2 < 0.01$. As the expression prediction accuracy is mainly determined by the sample size of the expression reference panel when using SUMMIT, we varied the sample size expression reference to obtain different prediction models with varying prediction accuracy. We further evaluated how the improvement in expression prediction accuracy affects TWAS power by conducting TWAS using these prediction models with different prediction accuracy. Briefly, we simulated gene expression levels E_g and phenotype values Y similar as before, set phenotypic heritability $h_p^2 = 0.2$, the expression heritability $h_e^2 = 0.01$ or $h_e^2 = 0.005$, and varied the sample size of expression reference panel n from (300, 600, 3000, 10000, 31684). For each n , we ran simulations for 1,000 times. We have shown that as the sample size of the reference panel increases, both prediction accuracy and subsequent TWAS power improve. These simulation results also confirmed that we should have reasonable power to detect the genes with small heritability. We have added the results to Figure 2.

We agree that including imputation models with testing $R^2 < 0.01$ would increase the burden of multiple testing. To study this, we evaluated SUMMIT’s performance for genes with $R^2 \geq 0.01$ under a less stringent p -value cutoff (due to models with $R^2 < 0.01$ excluded). We have confirmed that that the differences in p -value cutoff only have negligible impact on the SUMMIT in our real data analyses (Supplementary Figure 9). For example, SUMMIT identified 3,759 gene-trait associations for genes with $R^2 \geq 0.01$ using the less stringent cutoff and identified 3,685 gene-trait associations for genes with $R^2 \geq 0.01$ when using the more stringent cutoff. We have added discussion on Page 11.

Comment: *“The application of the method identified more significant genes. Explanation about why these genes were only identified by SUMMIT would be helpful for the readers to understand where the gain of power comes from. Most significant genes identified by SUMMIT and not by other method has $R^2 > 0.01$, is it because Predixcan or TWAS’s model is not as accurate? I think simulations regarding TWAS power for $R^2 > 0.01$ genes may help to answer this.”*

Reply: Thank you for your helpful comments. Yes, SUMMIT identified more genes mainly

because the accuracy improvement of expression prediction models leads to subsequent TWAS power gain. We have added the settings with $h_e^2 \in (0.005, 0.01, 0.1)$ (Figure 2c) and clearly shown that for all three scenarios, SUMMIT provides much more accurate expression prediction models and higher statistical power when a large expression panel is used. As the competing methods use an individual-level expression panel with a small sample size to build expression panel, they may not have enough power to identify some associations as demonstrated in Figure 2c. This also highlights the importance of using a large expression reference panel. The results were similar when $h_e = 0.05$ (Supplementary Figure 7). As a remark, the main difference between SUMMIT and PrediXcan or TWAS-Fusion is that SUMMIT uses summary-level expression reference panel while PrediXcan or TWAS-Fusion uses individual-level expression reference panel. We have added discussions on Page 7.

Comment: *“Another information that should be included in the manuscript is the connection with current method. It seems to me the impute expression method is similar/same to the “lasso sum” method developed for calculating PRS (Timothy Shin Heng Mak et al, 2017 Genetic Epidemiology), except that the method used here has been further extended it to use other penalties in addition to the lasso penalty. I think this point should be made clear in the manuscript and give proper credit to the lasso sum method. Another point is that this is not the first time that a summary level eQTL data be used for TWAS, a recent method has been published (Yang, Yi et al. “CoMM-S4: A Collaborative Mixed Model Using Summary-Level eQTL and GWAS Datasets in Transcriptome-Wide Association Studies.” Frontiers in genetics vol. 12 704538. 20 Sep. 2021, doi:10.3389/fgene.2021.704538”). I think current method has the benefit of easy implementation and integration with popular TWAS framework. But proper reference should be given to previous publications.”*

Reply: Thank you for your insightful suggestions. Yes, our method SUMMIT shares similarity with Lassosum as they both constructed the primary loss function using penalized regression under a summary-level data framework. Our method is partially motivated by Lassosum and we now discuss its connections in the revision. We have confirmed that Lassosum can be equally applied to eQTL summary data. However, SUMMIT achieved better performance in terms of prediction accuracy and subsequent statistical power in both simulations (Figure 2 on Page 6) and real data analyses (Figure 3 on Page 10). This is because SUMMIT makes two improvements. First, SUMMIT adds an additional step to

estimate the LD matrix by utilizing genetic distance information. Second, Lassosum only uses LASSO penalty, while SUMMIT considers five different types of penalties. We have added discussions on Page 14.

In addition, we thank you for pointing out “CoMM-S4” and we have acknowledged this paper on Page 14.

Comment: *“Minor point: it is not necessary to have subtitles for the figures, relevant information should be provided in figure legend.”*

Reply: We appreciate your inputs and have revised figures as suggested.

References

- [1] Cramer, J. S. (1987). Mean and variance of R^2 in small and moderate samples. *Journal of Econometrics*, 35(2-3):253–266.
- [2] Zhou, D., Jiang, Y., Zhong, X., Cox, N. J., Liu, C., and Gamazon, E. R. (2020). A unified framework for joint-tissue transcriptome-wide association and Mendelian randomization analysis. *Nature Genetics*, 52(11):1239–1246.

REVIEWER COMMENTS

Reviewer #1 (Remarks to the Author):

I commend the authors on their thorough responses to my prior suggestions. I have no additional comments.

Reviewer #2 (Remarks to the Author):

Comments have been properly addressed